# Crosstalk of Mast Cells and Natural Killer Cells with Neurons in Chemotherapy-Induced Peripheral Neuropathy

**DOI:** 10.3390/ijms241612543

**Published:** 2023-08-08

**Authors:** Hyun Don Yun, Yugal Goel, Kalpna Gupta

**Affiliations:** 1Hematology, Oncology, Veterans Affairs Long Beach Healthcare System, Long Beach, CA 90822, USA; 2Division of Hematology, Oncology, Department of Medicine, School of Medicine, University of California, Irvine, CA 92617, USA; ygoel@hs.uci.edu (Y.G.); kalpnag@hs.uci.edu (K.G.)

**Keywords:** natural killer cells, mast cells, proinflammatory cytokines, TNFα, IL-1β, IL-6, chemotherapy-induced peripheral neuropathy

## Abstract

Chemotherapy-induced peripheral neuropathy (CIPN) is a major comorbidity of cancer. Multiple clinical interventions have been studied to effectively treat CIPN, but the results have been disappointing, with no or little efficacy. Hence, understanding the pathophysiology of CIPN is critical to improving the quality of life and clinical outcomes of cancer patients. Although various mechanisms of CIPN have been described in neuropathic anti-cancer agents, the neuroinflammatory process involving cytotoxic/proinflammatory immune cells remains underexamined. While mast cells (MCs) and natural killer (NK) cells are the key innate immune compartments implicated in the pathogenesis of peripheral neuropathy, their role in CIPN has remained under-appreciated. Moreover, the biology of proinflammatory cytokines associated with MCs and NK cells in CIPN is particularly under-evaluated. In this review, we will focus on the interactions between MCs, NK cells, and neuronal structure and their communications via proinflammatory cytokines, including TNFα, IL-1β, and IL-6, in peripheral neuropathy in association with tumor immunology. This review will help lay the foundation to investigate MCs, NK cells, and cytokines to advance future therapeutic strategies for CIPN.

## 1. Introduction

Chemotherapy-induced peripheral neuropathy (CIPN) is a major comorbidity of cancer. The prevalence of CIPN is estimated at 30–40% of patients treated with neurotoxic agents [1]. Common chemotherapeutic drugs causing CIPN include proteosome inhibitors (e.g., bortezomib) [2], taxanes (e.g., paclitaxel) [3], platinum compounds (e.g., cisplatin) [4], and vinca alkaloids (e.g., vincristine) [5]. In addition, there has been increasing literature data reporting CIPN induced by newer classes of anticancer drugs such as brentuximab vedotin [6] and immune checkpoint inhibitors [7,8]. For example, brentuximab-induced neuropathy was reported in 56% of Hodgkin lymphoma patients, and a meta-analysis of twenty-three clinical trials reported that 4.2% of patients receiving an immune checkpoint inhibitor develop peripheral neuropathy [9]. Unfortunately, CIPN is often the dose-limiting factor of anticancer treatment, which often compromises the chemotherapeutic efficacy, resulting in poor cancer-related clinical outcomes [10,11]. Conversely, the improved survival of cancer patients with advances in cancer therapeutics intensifies the healthcare burden of CIPN [12]. Hence, developing therapeutic strategies for CIPN is critical to improving the quality of life of cancer survivors as well as the outcomes of cancer survival by enhancing tolerability to chemotherapeutics. There have been multiple clinical studies to advance therapeutic interventions including exercise, acupuncture, vitamins, minerals, antidepressants, topical agents, and gabapentinoids for CIPN, but the results have been disappointing with no or limited efficacy [13,14]. For example, gabapentin was not only ineffective in CIPN treatment with little improvement in quality of life [15], but the increased risks of falls and fractures associated with its use have been consistently reported [16,17].

The peripheral sensory network is a highly complex system with multiple non-sensory networks, including immune and non-immune cells such as epidermal keratinocytes, in cross-talk with sensory nerve endings in the skin [18]. An in vitro study using a coculture system of keratinocytes with sensory neurons has demonstrated that keratinocyte membranes directly depolarized by mechanical stimulation can propagate inward currents in the adjacent sensory neurons [19,20]. Keratinocytes are a major inadvertent target of chemotherapeutics, including immune checkpoint inhibitors [21], and cutaneous toxicities are known to be the most common adverse effect of checkpoint inhibitors [22]. Lichenoid dermatitis with dyskeratotic keratinocytes [23] and Steven-Johnson syndrome with apoptotic/necrotic keratinocytes [24] have been reported in patients treated with checkpoint inhibitors blocking CTLA-4, PD-1, and PD-L1, indicating that the unfitness of keratinocytes can be an important mechanism of immunotherapy-induced CIPN. 

Although various mechanisms of CIPN have been described in neuropathy-causing anticancer agents, the common pathway involved in CIPN is tightly interdigitated with immunological underpinnings where cytotoxic cells of innate immunity play a key role in propagation of neuroinflammation via neuro-immune synapse formation and release of proinflammatory cytokines [7,13,25]. Although innate immune cells such as mast cells (MCs) and natural killer (NK) cells are profoundly implicated in the pathogenesis of peripheral neuropathy, their roles in CIPN have remained under-appreciated. Moreover, as the current paradigm of oncological therapeutics has shifted from cytotoxic agents to cancer immunotherapy along with targeted therapy [26,27], MCs and NK cells have emerged as the key cellular components in cancer treatment responses and clinical outcomes [1,28,29,30,31]. In this review, we will focus on the interactions between mast cells, NK cells, and neuronal structure in peripheral neuropathy and their association with tumor immunology. This review will help lay the foundation to investigate MCs and NK cells in order to advance future therapeutic strategies for CIPN. 

Cytokines are the main humoral vehicle of communication between immune cells, including MCs and NK cells, and neurons. Particularly, proinflammatory cytokines such as TNFα, IL-1β, and IL-6 are well known to play a pivotal role in the progression of CIPN by inducing sensitization of nociceptors and axonal mitochondrial dysfunction and generating reactive oxygen stress in a neuroimmune environment [32]. In this review, we will focus on the interactions between MCs, NK cells, and neural structure and the role of cytokines and soluble factors in peripheral neuropathy in association with tumor immunology. This review will help lay the foundation to investigate MCs, NK cells, and cytokines to advance future therapeutic strategies for CIPN.

## 2. Mast Cells

Mast cells (MCs) are a part of the innate immune system, acting as the first and fastest responders to pathogens and allergens, and are distributed as tissue-resident myeloid cells primarily in body surfaces exposed to the external environment, including skin tissue and mucosa of the gastrointestinal tract [33,34]. MCs play an important role in both the tumor microenvironment (TME) and the CIPN. MCs are one of the primary innate immune cells in TME, attracted by stem cell factor (SCF) released by tumor cells [35]. MCs orchestrate proinflammatory immune responses by recruiting neutrophils, macrophages, eosinophils, T and B cells, and releasing a variety of inflammatory cytokines, including IL-1, IL-4, IL-6, and TNFα [36]. Moreover, MCs are an important source of angiogenic cytokines such as IL-8, VEGF, and FGF-2 and facilitate tumor invasion and metastasis by producing matrix metalloproteinases (e.g., MMP-2, MMP-9) [28]. Morphine, a narcotic substance commonly used for cancer-related pain, induces tumor progression by increasing tumor angiogenesis and mast cell activation, leading to poorer survival in a transgenic mouse model with breast cancer [37,38]. Recently, MCs in TME were reported to be a detrimental factor for treatment with a checkpoint inhibitor [29]. The presence of intratumoral MCs was associated with a poor immune response to anti-PD-1 therapy, and depletion of MCs by imatinib or sunitinib restored the efficacy of anti-PD-1 therapy, resulting in complete tumor regression in a murine melanoma model [29]. 

Similar to the detrimental effects of MCs in TME, inflammation induced by MCs plays an important role in peripheral neuropathy [33,39,40]. MCs in the epidermis are colocalized with the nerve terminals of unmyelinated small-diameter C-fibers and myelinated A-delta fibers, key neurons that convey pain stimulation to the trigeminal and dorsal root ganglions and brain [33,41]. The role of MCs in neuropathic pain is well described in a robust murine model of sickle cell anemia, a prototypical disease of severe pain episodes induced by inflammation, oxidative stress, and ischemia-reperfusion injury [42]. Vincent et al. reported that MC inhibition by treatment with cromolyn sodium or imatinib significantly reduces neuroinflammation, and genetic depletion of MCs in sickle cell mice attenuates chronic hypoxia-induced hyperalgesia [43]. Additionally, cannabinoids can alleviate neurogenic inflammation and hyperalgesia while mitigating mast cell activation in sickle cell mice [44], again highlighting the important role of mast cells in painful neuropathy.

MCs excrete or degranulate multiple substances that mediate communication with the neural system. Histamine, an inflammatory substance from MC degranulation, stimulates C fibers to release substance P (SP) [45,46], which is also directly released from MCs and activates adjacent nerve fibers [33]. Conversely, calcitonin gene-related peptide (CGRP), an algogenic substance released from sensory nerves, induces MCs to produce histamine [47]. Not surprisingly, the administration of antihistamine agents is well established to treat and prevent CIPN in clinical practice [48]. Antihistamines along with high-dose dexamethasone are also proven to be efficacious in preventing hypersensitivity (mast cell-mediated acute infusion reactions) induced by oxaliplatin, a platinum agent [49]. Tryptase, a serine proteinase, released from MC granules activates proteinase-activated receptor 2 (PAR-2) [50] that induces neurokinin-1 receptor-dependent hyperalgia [51]. Activation of PAR-2 by tryptase further stimulates afferent neurons to release proinflammatory neuropeptides, including CGRP and SP [52]. (Figure 1A,B) In a murine model, thermal hyperalgesia and tactile allodynia induced by repeated paclitaxel administration correlated with mast cell tryptase activity in peripheral tissue [53]. Treatment with FSLLRY-NH2, a PAR antagonist, or blocking PAR2 downstream signaling, including PLCβ, PKCε, and PKA, diminished paclitaxel-induced neuropathic pain [53]. Another important algogenic mediator released from MCs is sphingosine-1-phosphate (S1P). S1P is the product of sphingosine catalyzed by sphingosine kinases (SphK), activated by crosslinking of FcεRI, IgE receptor [54,55]. S1P binds to its receptors S1P1 and S1P2 in an autocrine manner [55]. S1P mediates mast cell degranulation and migration toward antigens. Furthermore, the pharmacological blockade of S1P on S1P1 mitigated cancer-induced bone pain and neuropathy in a murine model [56], indicating that the S1P pathway can be a major target for CIPN treatment. Additionally, a murine model demonstrated that fingolimod, an S1P1 modulator, attenuates paclitaxel- and oxaliplatin- induced neuropathy and reduces neuroinflammation [57]. 

## 3. Natural Killer (NK) Cells

NK cells are cytotoxic lymphocytes of innate immunity. Unlike T cells, lymphocytes of adaptive immunity, NK cells do not require major histocompatibility complex (MHC) restriction for target killing [58]. The interaction between inhibitory killer immunoglobulin-like receptors (KIR) on NK cells and HLA class I molecules on target cells generates inhibitory signals to NK cells via tandem immunoreceptor tyrosine-based inhibitory motifs (ITIMs) [31,59]. NK cells are educated by KIR-HLA interaction, which leads to enhanced NK cell cytotoxicity against foreign, malignant, or virally transformed cells lacking the normal expression of HLA class I (i.e., “missing self”) [60]. The importance of KIR-HLA genotype in the clinical outcomes of cancer treatment has been well described in allogeneic hematopoietic stem cell transplantation (allo-HCT). The KIR genotype can be simply categorized into haplotypes A and B according to the gene contents within the haplotypes. Donors with KIR haplotype B significantly improve disease-free survival in patients with acute myeloid leukemia [61,62,63] and non-Hodgkin lymphoma [64]. Moreover, AML patients’ HLA-C haplotypes confer a significant clinical benefit in allo-HCT [65]. Another potent mechanism of NK cytotoxicity is antibody-dependent cell-mediated cytotoxicity (ADCC). Monoclonal antibodies (moAb) bind to antigens expressed on the surface of tumor cells via the antigen-binding portion (Fab fragment). The other end of moAb is the Fc portion that is recognized FcγRIIIA/CD16a on NK cells. The ligation of CD16a with the Fc portion of the moAb generates potent activating signals to NK cells via the immunoreceptor tyrosine-based activation motif (ITAM) [66,67]. The development of moAb has revolutionized the landscape of cancer treatment [68]. Novel molecules such as bispecific or trispecific killer engagers (i.e., BiKE, TriKE) demonstrated promising antitumor activities by harnessing the ADCC of NK cells [30,69,70,71]. Disintegrin and metalloprotease-17 (ADAM17) expressed on activated NK cells cleaves CD16, which in turn attenuates ADCC activity in NK cells [72]. Inhibition of ADAM17 to prevent CD16 shedding significantly enhanced ADCC mediated by rituximab, a monoclonal anti-CD20 antibody against CD20-expressing tumor cells [72], which indicates that ADCC of NK cells plays a pivotal role in therapeutic efficacy of monoclonal antibodies in cancer treatment. Lastly, the NKG2D pathway is another important mechanism of NK cell activation. Ligands for NKG2D, an activating NK receptor, are often expressed by tumor cells, transformed cells, or infected cells [73]. NKG2D ligands include the RAE (α–ε) encoded by Raet1 genes, H60 (a–c), and MULT1 families in mice and MICA/MICB and ULBPs (1–6) in humans [74,75]. The NKG2D signaling pathway is regarded as the primary host defense mechanism for eradicating “dangerous” cells, as NKG2D ligands can be overexpressed in malignant or infected cells that are subsequently eliminated by NK cells [74]. Shedding MICA/B on tumor cell surfaces and subsequent soluble MICA/B can downregulate NKG2D expression on NK cells, which promotes tumor immune escape by impairing NK cell antitumor activity [76,77]. On the other hand, prolonged activation of NK cells by NKG2D signals results in NK cell exhaustion [78], another potential mechanism of tumor immune evasion. Besides, NK cell cytotoxicity is further determined by the overall balance between inhibitory and activating signals generated by NK cell receptors and ligands expressed on target cells or from soluble factors [31,79,80].

The effect of NK cell cytotoxicity on neuronal degeneration has been described in multiple murine models. In 1982, chronic administration of guanethidine, an adrenergic blocking agent, caused extensive destruction of sympathetic nerves and resulted in “small cell” infiltration [81]. Pretreatment with immunosuppressive agents or irradiation effectively protected against neuronal destruction, indicating that neuronal destruction is mediated by immune mechanisms [81]. Guanethidine administration induced neuronal destruction by infiltration of mononuclear cells even in athymic nude rats, indicating T cell-independent immune-mediated neuronal destruction [82]. A subsequent study revealed that syngeneic IL-2-activated NK cells directly killed embryonal dorsal root ganglia (DRG) neurons via perforin-dependent cytotoxicity, whereas NK cell-mediated lysis was not observed in hippocampal neurons [83]. RAE-1 (an NKG2D ligand) was selectively expressed in embryonal DRG neurons, and anti-NKG2D monoclonal antibodies impaired NK cell-mediated destruction of DRG neurons, which indicates that NKG2D-dependent NK cell cytotoxicity contributes to the degeneration of DRG neurons. NKG2D-dependent NK cell destruction of injured peripheral nerves has been well described [84]. In contrast to embryonal DRG neurons, adult DRG neurons do not express RAE1 as much [84]. However, following axonal injury of peripheral nerves, RAE1 is re-expressed in adult DRG cells, allowing augmented cytotoxicity by activated NK cells via the NKG2D signaling pathway [84]. Although the increased content of granzyme B released from NK cells was identified in injured peripheral nerves, the mere presence of granzyme B in in vitro culture media by separating NK cells from embryonic DRG neurons did not further degenerate the DRG neurons [84]. Hence, the neuro-immune synapse formation of NK cells with nerve fibers is critical for NK cell-induced nerve degeneration. Interestingly, in vivo endogenous NK cytotoxic responses to crushed nerve injury paradoxically reduced chronic neuropathic hypersensitivity in a murine model via the clearance of partially injured sensory axons by activated NK cells [84]. (Figure 1D) Alleviation of neuropathic pain with further degeneration of partially injured exons by NK cytotoxicity is supported by another murine study, where partial sensory fiber loss induced hyperalgesia but more severe axonal loss mitigated hypersensitivity responses [85]. In another murine model of neuropathic pain, electroacupuncture, an effective treatment for neuropathic pain, increased NK cell percentages in the spleen and peripheral blood and NK cell activity measured by the methyl thiazolyl tetrazolium (MTT) assay [86]. However, electroacupuncture treatment did not induce the analgesic effect in mice with in vivo depletion of NK cells, suggesting that NK cells play an important role in the treatment of neuropathic pain [86]. A prospective cohort study reported a significant inverse correlation between the frequency of NK cells in CSF and mechanical pain sensitivity in patients with herpes zoster neuralgia and polyneuropathy (P = 0.004, r = −0.596), indicating a protective role of NK cells in chronic neuropathy [87]. Moreover, the severity of neuropathy inversely correlates with NK cell numbers and NK cell-specific transcription levels in the peripheral blood of patients with chronic inflammatory demyelinating polyneuropathy, again implying the protective role of NK cells in neuropathy [88].

## 4. Communication between Mast Cells and NK Cells

There is a paucity of data on the interaction between MCs and NK cells specifically in CIPN, although both immune cell groups play significant roles in peripheral neuropathy as described above. However, the communication between MCs and NK cells has been described in other clinical contexts. MCs are shown to attract NK cells via the production of chemokines and cytokines. For example, MCs recruit NK cells to enhance viral clearance during dengue infection [89]. MCs produce CXCL8 in response to reovirus, resulting in NK cell chemotaxis [90] and activating NK cells via the type I interferon response [91]. On the other hand, a hepatocarcinoma murine model demonstrated that tumor-infiltrating mast cells activated by tumor-derived stem cell factor (SCF) augment immunosuppression, and adenosine released by MCs suppresses NK cell activity with the reduction of interferon gamma release in TME [35].

Although NK cells are potent cytotoxic lymphocytes, MCs seem resistant to NK cell cytotoxicity [69,92]. Yun et al. demonstrated that NK cells can successfully irradiate MCs through enhanced ADCC in an in vitro NK cytotoxicity assay [69]. Trispecific killer engager (TriKE), a construct combining a single chain variable fragment (scFv) against CD33 highly expressed on the surface of MCs [93], a scFv against CD16 expressed on NK cells, and IL-15, a key cytokine for NK cell survival, activation, and proliferation, inserted in between as a linker (termed 161533 TriKE), was used to target MCs in this study [69]. 161533 TriKE potently induced NK cell cytotoxicity against MCs, indicating the great therapeutic potential of 161533 TriKE to target MCs by augmenting NK ADCC [69]. (Figure 1C) Investigating the therapeutic potential of 161533 TriKE is warranted in mast cell-associated peripheral neuropathy, especially in the context of CIPN, as NK cell activity is tightly linked to cancer control as well as peripheral neuropathy.

## 5. Proinflammatory Cytokines

Treatment with cytotoxic chemotherapeutic agents results in cell deaths in neoplastic and normal tissue, which leads to systemic tissue damage and proinflammatory cytokine releases by immune cells (Figure 2) [94,95]. In addition, chemotherapy can directly induce inflammatory responses in the neuronal structure (e.g., dorsal root ganglia) by augmenting the expression of proinflammatory cytokines such as TNFα [96].

As chronic inflammation is a key risk factor for developing various malignancies, inflammatory mediators, including IL-6 and TNFα are known to be involved in carcinogenesis and cancer progression (Figure 2) [97]. Proinflammatory cytokines contribute to generating reactive oxygen species (ROS), and the level of ROS highly correlates with proinflammatory cytokines including TNFα, IL-6, and IL-1β [98,99]. TNFα promotes tumor progression by inducing ROS via cytosolic phospholipase A(2) [100], causing DNA damage, a major carcinogenesis process [101]. Aquaporin (AQP)-3 and AQP-5-mediated diffusion of H_2_O_2_, a prototypic ROS molecule, facilitates pancreatic cancer cell migration [102], which characterizes cancer invasion and metastasis. In addition, proinflammatory cytokines serve as growth factors for various types of cancer. For example, metastatic tumor growth can be induced by TNFα release from host hematopoietic cells that mediates NF-κB activation in tumor cells [103]. Activation of the IL-6/STAT3 signaling pathway promotes tumor metastasis [104], and IL-17 released from CD8+ T cells also stimulates cutaneous tumor growth [105]. Multiple in vitro studies have demonstrated that the process of epithelial-to-mesenchymal transition (EMT), a pivotal step for cancer progression with tumor invasion and distant metastasis, is activated by proinflammatory cytokines including TNFα, IL-6, IL-8, and IL-1β [106,107,108,109,110]. ROS also contributes to the EMT process [111], which can be another potential mechanism of proinflammatory cytokine-induced EMT. The protumorigenic effect of inflammation can be mediated by the promotion of angiogenesis, the process of new blood vessel formation to supply oxygen and nutrients to the malignant tissue. Vascular endothelial growth factor (VEGF) is the key molecule for angiogenesis secreted by cancer cells [112]. IL-6-induced STAT3 phosphorylation is associated with increased expression of VEGF and VEGFR2 [113,114]. Moreover, tumor angiogenesis was completely abrogated by a TNFα-neutralizing antibody, indicating that TNFα promotes tumor angiogenesis [115]. 

MCs are one of the major sources of proinflammatory cytokines. Activated mouse MCs by stem cell factor (SCF) via an interaction with c-kit highly express and release IL-6 [116,117]. MCs produce IL-1 when activated, whereas TNFα is stored in MCs as a preformed mediator [118]. Moreover, MCs can stimulate macrophages to produce IL-1β, as seen in rheumatoid arthritis [119]. Conversely, proinflammatory cytokines affect the function and development of MCs. Based on a murine model, IL-6 and TNFα may contribute to the development of MCs from mast cell precursors [120]. In addition, IL-6 promotes MC survival [121,122]. Under hypoxic conditions, treatment with neutralizing anti-IL-6 antibodies compromised mast cell survival [123]. IL-1, another proinflammatory cytokine, induces IgE-activated MCs to release IL-6 and TNFα along with Th2-related cytokines [124].

Like MCs, NK cell function is tightly associated with proinflammatory cytokines (Figure 2). It is well known that NK cell cytotoxicity is compromised in hyperinflammatory conditions including hemophagocytic lymphohistiocytosis [125], juvenile rheumatoid arthritis, and macrophage activation syndrome [126]. IL-6, primary proinflammatory cytokine implicated in hyperinflammation and cytokine release syndrome [127], impairs NK cell cytotoxicity by downregulating the expression of cytotoxic granules, including perforin and granzyme B [128]. NK cell cytotoxicity can be mediated by TNFα, or TNF-related apoptosis-inducing ligand (TRAIL), a type II membrane protein with homology to TNF [129], which generates apoptotic signaling in the target cells [130,131]. However, TNFα can mediate detrimental effects on NK cell function and survival. Endogenous TNFα induces functional anergy and apoptosis of NK cells activated by triggering CD16 signaling [132]. Proinflammatory cytokines can indirectly compromise NK cell function through ROS induction as well. In an in vitro study, NK cell cytotoxicity was inversely correlated with intracellular ROS production in tumor cells [133]. Moreover, phagocyte-derived ROS impairs NK cell function by diminishing the expression of the NKp46 natural cytotoxicity receptor and NKG2D, an NK cell-activating receptor [134]. ROS-induced NK cell dysfunction can be mediated by A Disintegrin and Metalloprotease 17 (ADAM17), also known as TNFα-converting enzyme (TACE). Oxidative stress and mitochondrial ROS contribute to the increased activity of ADAM17 [135,136]. ADAM17 expressed on NK cells cleaves CD16, leading to the shedding of CD16, the key molecule that mediates ADCC by crosslinking [72,137]. Hence, ROS generated by inflammation can diminish NK cell function by impairing ADCC of NK cells as well as downregulating NK cell-activating receptors. 

Neuroinflammation is a common denominator in CIPN, where proinflammatory cytokines serve as messengers for neuro-immune communication [32]. Injury to nervous tissue induces denervated Schwann cells to mount myelomonocytic responses by chemoattraction mediated via leukemia inhibitory factor (LIF) and monocyte chemoattractant protein-1 (MCP-1) in an IL-6-dependent manner [138]. Moreover, activated glial cells following nerve damage produce multiple proinflammatory cytokines, including TNFα, IL-1β, and IL-6 [139,140]. Proinflammatory cytokines not only attract immune cells to the neuroinflammatory tissue but also directly sensitize nociceptors (Figure 2). For example, TNFα and IL-1β can stimulate A- and C- fibers [141]. In rat DRG neurons, the expression of TNF receptors (TNFR1 and TNFR2), TNF-activated p38 mitogen-activated protein kinase (p38MAPK), and c-jun N-terminal kinase (JNC) were observed in immunocytochemical, analysis while TNF-evoked transient increases in [Ca2+] were detected [142]. In a murine model, subcutaneous injection of TNFα resulted in mechanical sensitivity in C nociceptors in a dose-dependent manner, accounting for the generation of hyperalgesia in inflammation [143]. Furthermore, the plantar administration of IL-1β induced a hypersensitive cutaneous reaction to mechanical stimulation [144]. IL-1 signaling impairment significantly reduced pain sensitivity to mechanical and thermal stimulation [145]. Not surprisingly, epidural administration of neutralizing anti-TNFα and anti-IL-1β antibodies markedly diminished pain sensitivity in an additive manner [146]. Although the data had been conflicting, there are multiple clinical studies demonstrating that blockade of proinflammatory cytokines alleviates the symptoms of peripheral neuropathy. A retrospective study demonstrated that perispinal administration of etanercept, a TNFα inhibitor, significantly reduced pain, sensory disturbance, and weakness in patients with treatment-refractory back and neck pain at 1 week, 2 weeks, and 1 month after the treatment [147]. In a randomized, double-blind, placebo-controlled study, the epidural administration of etanercept resulted in significant symptom improvement in patients with sciatica, although this study was limited by a small sample size (n = 24) [148]. In addition to TNFα, targeting IL-6 has been studied in humans. A prospective study comparing the efficacy of epidural administration of tocilizumab, an anti-IL-6 receptor monoclonal antibody, with dexamethasone treatment revealed that tocilizumab treatment was significantly more efficacious for symptom alleviation in patients with lumbar spinal stenosis [149]. 

Another mechanism by which proinflammatory cytokines trigger neuropathy is the generation of ROS, the main byproduct of proinflammatory cytokines as well as cancer cells [150] and the tumor microenvironment [151]. In fact, chemotherapeutic agents are highly potent in generating ROS, causing multiple tissue damages [152]. The peripheral nervous system is regarded as particularly susceptible to oxidative stress [153]. Naturally, ROS is associated with the development and maintenance of peripheral neuropathy. For example, Mitochondrial ROS production was markedly increased in neuropathic dorsal horns, suggesting that increased neuronal ROS production may be involved in neuropathic sensitization [154], and increased spinal ROS levels due to the production of superoxide from the mitochondria of dorsal horn neurons are associated with maintaining capsaicin-induced hypueralgesia [155,156]. Moreover, inhibition of ROS-by-ROS scavengers including N-tert-Butyl-α-phenylnitrone (PBN) and 4-hydroxy-2,2,6,6-tetramethylpiperidine-1-oxyl (TEMPOL) markedly reduced paclitaxel-induced painful peripheral neuropathy, indicating that ROS plays a central role in the pathogenesis of CIPN [157]. Interestingly, chemotherapy treatment induces an ROS-dependent DNA damage response, which results in upregulation of NK cell ligands on the target cells, leading to NK cell activation [158]. As NK cell-cytotoxicity can exert beneficial therapeutic effects for peripheral neuropathy [84], NK cell-directed treatment can be especially effective in ROS generation-associated CIPN. 

It is not surprising that anti-inflammatory cytokines counterbalance proinflammatory cytokines and are associated with beneficial effects in peripheral neuropathy. For example, intrathecal administration of plasmid DNA encoding IL-10 prevented and alleviated paclitaxel-induced peripheral neuropathy in a murine model [159]. Notably, this IL-10-gene therapy resulted in significant reduction in paclitaxel-induced mRNA expression of IL-1β and TNFα in the lumbar DRG, indicating a pivotal role of proinflammatory cytokines in eliciting CIPN. Transforming growth factor β (TGFβ) is another anti-inflammatory cytokine associated with a reduction in neuropathic pain. A murine model demonstrated that intrathecal treatment of TGFβ significantly reduced neuropathic pain by inhibiting the activation of spinal microglia and astrocytes and mitigating spinal inflammatory responses to nerve injury [160]. TGFβ treatment also reduced the expression of IL-1β and IL-6 in the spinal cord with peripheral nerve injury [160]. 

## 6. Conclusions

CIPN is a major challenge in cancer treatment. However, the efficacy of therapeutic interventions currently available for CIPN treatment is suboptimal. The biology of CIPN is highly complex. Although MCs and NK cells are known to be highly implicated in the pathogenesis of peripheral neuropathy, there is a paucity of studies on the pathobiology of MCs and NK cells in CIPN. Moreover, the biology of proinflammatory cytokines associated with MCs and NK cells in CIPN is particularly under-evaluated. Based upon the current data, targeting mast cells, proinflammatory cytokines, and/or augmenting the NK cell function in the neuro-immune microenvironment has the potential to improve CIPN. Hence, further studies on the biology of mast cells, NK cells, and their interactions through proinflammatory cytokines in CIPN are warranted. 

## Figures and Tables

**Figure 1 ijms-24-12543-f001:**
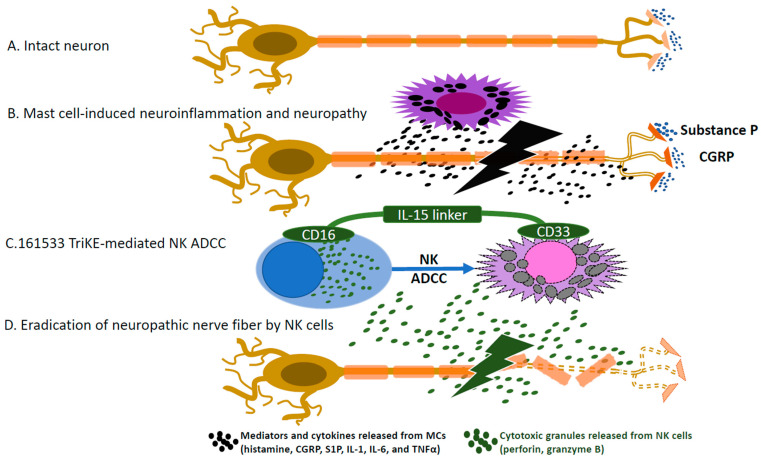
The potential role of mast cells (MCs) and NK cells in the pathobiology of CIPN. (**A**). Intact neuron prior to insults by mast cells (**B**). Mast cells induce neuroinflammation by MC release of mediators and neuropeptides, which in turn mediate releases of algogenic substances from neuronal endings. (**C**). 161533 TriKE augments ADCC of NK cells against CD33 expressing MCs. (**D**). NK cells release cytotoxic granules, which results in complete loss of neuropathic nerve fibers, which can mitigate neuropathic symptoms.

**Figure 2 ijms-24-12543-f002:**
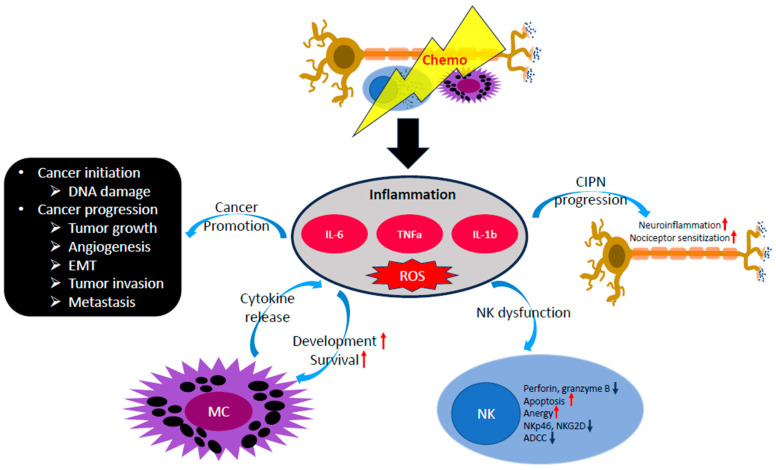
The impact of proinflammatory cytokines on chemotherapy-induced peripheral neuropathy (CIPN). Proinflammatory cytokines can exert negative multifaceted effects in CIPN by disease progression of cancer, activation and maintenance of mast cells (MCs), NK cell dysfunction, neuroinflammation and nociceptor sensitization.

## Data Availability

Not applicable.

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
