# Peer review of "Crosstalk of Mast Cells and Natural Killer Cells with Neurons in Chemotherapy-Induced Peripheral Neuropathy"

_ijms, 2023, doi:10.3390/ijms241612543_

Round 1

Reviewer 1 Report

Comments and Suggestions for Authors

CIPN  is a significant comorbidity in cancer. Mast cells (MCs) and NK cells are implicated in the CIPN but not clearly understood. In this review article, authors highlight the interactions between MCs, NK cells, and neuronal cells through inflammatory cytokines such as TNFα, IL-1β, and IL-20 6 in peripheral neuropathy-associated tumor immunology. Authors indicate that will initiate studies on MCs, NK cells, and cytokines to improve the treatment options for CIPN. This is an important article in the field of CIPN especially on the role of MCs and NK cells. It is a well-organized paper with the latest information in the field. Figure 1 clearly show potential role of MCs and NK cells in the pathogenesis of CIPN. However, some minor changes/addition may be useful for this paper as suggested below.

Figure 1B  - may indicate some main MCs-derived inflammatory mediators that act on neurons in the figure. May also put similar important molecules for NK cells on neurons.

Authors may add some info if available on the MC phenotypes in CIPN pathogenesis/treatment.

May include some info on NK cell subsets and the role of perforin and granzyme in CIPN pathogenesis/treatment if available.

Author Response

Response to Reviewer #1.

Thank you very much for your constructive and kind comments. Here are our responses to the reviewer’s comments.

  1. Figure 1B - may indicate some main MCs-derived inflammatory mediators that act on neurons in the figure. May also put similar important molecules for NK cells on neurons.

Authors: we appreciate the reviewer’s comment. Indicating and specifying the inflammatory mediators and cytokines released from mast cells (MCs), and cytotoxic granules released from NK cells will definitely help the readers to understand the effects of MCs and NK cells in chemotherapy-induced peripheral neuropathy (CIPN). According to the reviewer’s comment, we have revised figure 1 specifying the mediators and cytokines released from MCs and cytotoxic granules released from NK cells.

  1. Authors may add some info if available on the MC phenotypes in CIPN pathogenesis/treatment.

Authors: Thank you for the comment. Unfortunately, to our best knowledge, the phenotype and characteristics of MCs in the pathogenesis and progression of peripheral neuropathy, especially in CIPN are not yet well established.  However, it is highly valuable to characterize the phenotype of MCs associated with CIPN in order to identify therapeutic targets of CIPN.

  1. May include some info on NK cell subsets and the role of perforin and granzyme in CIPN pathogenesis/treatment if available.

Authors: We appreciate the reviewer’s highly insightful comment. CD56dim and CD56bright NK cell subsets behave quite differently: CD56dim NK subset is more cytotoxic with abundant release of perforin and granzyme whereas CD56bright NK cells release more proinflammatory cytokines such as interferon gamma [1]. Therefore, although there is no definitive published evidence for differential effects of NK cell subsets in CIPN pathogenesis, it is highly plausible to hypothesize that CD56dim NK cells will play a critical role in mitigating CIPN progression by the eradication of partially injured neuropathic nerves via NK cell cytotoxicity. We mentioned the effect of cytotoxic granules. For example, the death of embryonal dorsal root ganglion (DRG) neurons was observed by perforin-dependent cytotoxicity of IL-2-activated NK cells [2]. Moreover, increased release of granzyme B from NK cells was observed in peripheral nerve injury, but the mere presence of granzyme B did not induce DRG degeneration [3]. Unfortunately, it is hard to draw a definitive conclusion based on the scant published evidence yet. The role of cytotoxic NK cell granules including perforin and granzyme B deserve definitely deserves further investigation.  

References

  1. Cooper, M.A.; Fehniger, T.A.; Caligiuri, M.A. The Biology of Human Natural Killer-Cell Subsets. Trends Immunol. 2001, 22, 633–640, doi:10.1016/S1471-4906(01)02060-9.
  2. Backström, E.; Chambers, B.J.; Kristensson, K.; Ljunggren, H.G. Direct NK Cell-Mediated Lysis of Syngenic Dorsal Root Ganglia Neurons in Vitro. J. Immunol. Baltim. Md 1950 2000, 165, 4895–4900, doi:10.4049/jimmunol.165.9.4895.
  3. Davies, A.J.; Kim, H.W.; Gonzalez-Cano, R.; Choi, J.; Back, S.K.; Roh, S.E.; Johnson, E.; Gabriac, M.; Kim, M.-S.; Lee, J.; et al. Natural Killer Cells Degenerate Intact Sensory Afferents Following Nerve Injury. Cell 2019, 176, 716-728.e18, doi:10.1016/j.cell.2018.12.022.

Reviewer 2 Report

Comments and Suggestions for Authors

The paper sounds very important from a practical point of view. And hence can progress to the next step after the authors will revise the paper according to the following major comments: 

- The abstract section should be rewritten. This section should be the highlight of the research. The authors should present the aim of the research, the results, and the different methods to achieve these results in a very clear way. 

- Nomenclature should be added to the paper as well as mathematical\pgysical units of each variable and parameter. 

- It is very hard to follow the mathematical formulations without an extended explanation in detail.  

- In section 3 the authors present the results and after 2 lines they present a theorem. Please transfer all the theorems and their proofs to the previous section. 

- The following papers should be added to the current research: 

1: Nave, Op. (2020). Modification of Semi-Analytical Method Applied System of ODE. In Modern Applied Science (Vol. 14, Issue 6, p. 75). Canadian Center of Science and Education. https://doi.org/10.5539/mas.v14n6p75

2: BROADBRIDGE, P., & GOARD, J. (2023). EXACT SOLUTIONS OF HYPERBOLIC REACTION-DIFFUSION EQUATIONS IN TWO DIMENSIONS. In The ANZIAM Journal (pp. 1–17). Cambridge University Press (CUP). https://doi.org/10.1017/s1446181123000093

- The discussion section should be extended extensively.  and separate from the conclusion section. 

Author Response

Response to Reviewer #2.

Thank you very much for your constructive and kind comments. Here are our responses to the reviewer’s comments.

  1. The abstract section should be rewritten. This section should be the highlight of the research. The authors should present the aim of the research, the results, and the different methods to achieve these results in a very clear way.

Authors: thank you for your careful review. Unfortunately, we respectfully disagree with your comment on rewriting the abstract by presenting the sections on aim, results, and methods. The reviewer’s comment is more applicable to an original research paper. However, this is a comprehensive review paper for which a brief summary and highlight of the contents will be more appropriate as in our abstract.

  1. Nomenclature should be added to the paper as well as mathematical\pgysical units of each variable and parameter. It is very hard to follow the mathematical formulations without an extended explanation in detail.

Authors: we appreciate the reviewer’s concern. The authors discussed each other on these comments, and tried hard to address the concerns raised here. However, it was hard to exactly understand what the reviewer’s comments referred to. For example, the reviewer indicated the need for nomenclature for mathematical units of variables and parameters, but it is difficult to understand which units of variables or parameters the reviewer refers to. Moreover, we have not used mathematical formulations that need extended explanations in detail.

  1. In section 3 the authors present the results and after 2 lines they present a theorem. Please transfer all the theorems and their proofs to the previous section.

Authors: we appreciate the reviewer’s comment. Unfortunately, authors are unable to understand this comment. For example, it is difficult for authors to understand what “section 3” and “transfer all the theorems” the reviewer refers to as we did not use the term “Section” in our paper. 

  1. The following papers should be added to the current research: 1: Nave, Op. (2020). Modification of Semi-Analytical Method Applied System of ODE. In Modern Applied Science (Vol. 14, Issue 6, p. 75). Canadian Center of Science and Education. https://doi.org/10.5539/mas.v14n6p75. 2: BROADBRIDGE, P., & GOARD, J. (2023). EXACT SOLUTIONS OF HYPERBOLIC REACTION-DIFFUSION EQUATIONS IN TWO DIMENSIONS. In The ANZIAM Journal (pp. 1–17). Cambridge University Press (CUP). https://doi.org/10.1017/s1446181123000093

Authors: we appreciate the reviewer’s comment. Unfortunately, the papers that the reviewer recommended are completely irrelevant to our study and we cannot cite those papers.

  1. The discussion section should be extended extensively. and separate from the conclusion section.

Authors: we appreciate the reviewer’s comment. Unfortunately, our paper does not have a discussion section. Hence, the reviewer’s comment cannot be applicable to our current review paper.

Round 2

Reviewer 2 Report

Comments and Suggestions for Authors

Sorry but I can't see the changes that made by the authors. 

Please highlight the changes in the revised paper

Author Response

Thank you for your review. As we responded to your comments, there was no revision made according to your comments and the rationale has been written in our response. 

Sincerely yours,

Hyun Don Yun
